# Natural Patterns of Sitting, Standing and Stepping During and Outside Work—Differences between Habitual Users and Non-Users of Sit–Stand Workstations

**DOI:** 10.3390/ijerph17114075

**Published:** 2020-06-08

**Authors:** Lidewij R. Renaud, Maaike A. Huysmans, Hidde P. van der Ploeg, Erwin M. Speklé, Allard J. van der Beek

**Affiliations:** 1Department of Public and Occupational Health, Amsterdam Public Health Research Institute, Amsterdam UMC, Vrije Universiteit Amsterdam, 1081 BT Amsterdam, The Netherlands; l.renaud@amsterdamumc.nl (L.R.R.); hp.vanderploeg@amsterdamumc.nl (H.P.v.d.P.); erwin.spekle@arbounie.nl (E.M.S.); a.vanderbeek@amsterdamumc.nl (A.J.v.d.B.); 2Arbo Unie, Occupational Health Service, 3526 KS Utrecht, The Netherlands

**Keywords:** movement patterns, prolonged sitting, prolonged standing, office workers, sit–stand workstations, users and non-users, working and non-working hours

## Abstract

Sit–stand workstations have shown to reduce sitting time in office workers on a group level. However, movement behaviour patterns might differ between subgroups of workers. Therefore, the objective of this study was to examine sitting, standing and stepping outcomes between habitual users and non-users of sit–stand workstations. From an international office population based in the Netherlands, 24 users and 25 non-users of sit–stand workstations were included (all had long-term access to these workstations). Using the ActivPAL, sitting, standing and stepping were objectively measured during and outside working hours. Differences in outcomes between users and non-users were analysed using linear regression. During working hours, users sat less (−1.64; 95% IC= −2.27–−1.01 h/8 h workday) and stood more (1.51; 95% IC= 0.92–2.10 h/8 h workday) than non-users. Attenuated but similar differences were also found for total sitting time over the whole week. Furthermore, time in static standing bouts was relatively high for users during working hours (median= 0.56; IQR = 0.19−1.08 h/8 h workday). During non-working hours on workdays and during non-working days, no differences were found between users and non-users. During working hours, habitual users of their sit–stand workstation sat substantially less and stood proportionally more than non-users. No differences were observed outside working hours, leading to attenuated but similar differences in total sitting and standing time between users and non-users for total days. This indicated that the users of sit–stand workstations reduced their sitting time at work, but this seemed not to be accompanied by major carry-over or compensatory effects outside working hours.

## 1. Introduction

Office workers spend most of their time at the office in a sitting position [1]. Furniture that provides an alternative for sitting at the office, for instance sit–stand workstations, is gaining popularity. At sit–stand workstations, office workers can perform (computer) work tasks, while being able to alternate their posture between sitting and standing [2]. Interventions, such as the introduction of sit–stand workstations, have increasingly been evaluated on their effectiveness to reduce sitting time [3], their effect on productivity [4,5] and musculoskeletal symptoms [6]. A recent review, in which 53 studies (published between 2007 and 2017) on the effectiveness of sit–stand workstations were identified, concluded that sit–stand workstations were effective in reducing sitting time [7]. Reductions in sitting time, associated with the use of sit–stand workstations, were mainly replaced by increased standing times [8], also when multi-component interventions (aimed at decreasing sitting time and increasing physical activity) were implemented [9,10]. Sitting and standing at work in habitual users of sit–stand workstations were weakly and moderately correlated with total daily sitting and standing time, respectively [11], indicating that effects were mainly established during working hours.

At present, the rationale behind the implementation of sit–stand workstations is mainly driven by the evidence from the recently emerging research field of sedentary behaviour, which has shown that prolonged sitting, accumulated during the whole day, seems to be associated with several health risks in the long-term, including type II diabetes [12,13], cardiovascular disease [14,15] and premature mortality [16]. It has been indicated that breaks in sitting, interrupted with light intensity activity, could be beneficial for cardiovascular health [17]. Furthermore, using active workstations have shown to increase energy expenditure in individuals with overweight and obesity [18], and the use of sit–stand workstations may reduce the risk of lower back pain [19]. However, effects on biomarkers for cardio-metabolic health, including glucose and lipoproteins, remain ambiguous for sitting compared to standing and stepping [20,21,22] and for the use of sit–stand workstations [23]. It remains unclear if replacing sitting with standing is sufficient to reduce the health risks associated with large volumes of sitting time [24]. Therefore, scepticism has been expressed about the use of sit–stand workstation as a solution to reduce health risks [25]. Nonetheless, in practice, sit–stand workstations are widely perceived as a remedy against the health risks of prolonged sitting [26,27].

After the implementation of sit–stand workstations, long-term evaluations have shown that adherence to the use of the standing option decreased over time [28,29,30] or remained absent on a group level [31]. In general, participation levels in workplace health promotion programs, could be lower than 50% [32]. Furthermore, high baseline physical activity levels in participants induced ceiling effects for interventions aimed at increasing these levels [33], while adherence to physiotherapy treatment interventions (including physical activity) were lower in participants with low baseline levels of physical activity [34]. This indicates that the level of adherence to the use of sit–stand workstations during working hours, might differ between individuals with different movement behaviour patterns outside working hours. Office workers with predominantly sedentary occupations have shown to be more sedentary during working days and working hours compared to non-working days and non-working hours, respectively [35]. However, a profound understanding of movement behaviour patterns in employees with access to sit–stand workstations remains missing, especially when habitual (long-term) users and non-users are considered. 

Hence, it is important to study the movement behaviour patterns of habitual users and non-users of sit–stand workstations, during and outside working hours. In a previous survey-study, we identified three user groups of sit–stand workstations, including daily users, monthly/weekly users and non-users, within an office population with long-term access to the furniture [36]. In the current study, the research question was: Do users and non-users of the sit–stand workstations differ in movement behaviour outcomes during working and non-working hours and during total days, workdays and non-workdays? Movement behaviour outcomes included sitting time; standing time; stepping time; sit-to-stand transitions; sitting bouts of different durations; and static standing bouts. 

## 2. Materials and Methods 

### 2.1. Design 

This cross-sectional study was conducted at a semi-governmental international organisation, employing an office population with a variety of European nationalities, situated in the Netherlands. Data were collected in November and December 2017. Participants were selected from an earlier conducted survey study [36], in which participants (n = 1098 from a randomly drawn sample of employees within the organisation) had indicated their frequency of use of the sit–stand workstation. The data were collected using a seven full-day protocol. Written informed consent was obtained by the researcher prior to the measurements. The Medical Ethics Review Committee of the VU University Medical Center Amsterdam has approved this study (2016.346).

### 2.2. Participants

Inclusion criteria for participants in this study were: (1) indicated to be interested in further research and provided an email address in the previously conducted survey study; (2) being identified as a user (i.e., using the standing option at least daily) or as a non-user (i.e., using the standing option less than once per week) of the sit–stand workstation, and (3) being involved in the primary process of the organisation, which involved homogeneous (computer) work tasks. Exclusion criteria for participants in this study were: (1) any specific self-reported reason for not being able to use the desk, i.e., (temporarily) unable to stand for some time (being pregnant, having a disability, etc.); (2) using the workstation semi-frequently (i.e., more than once a week but less than once a day); (3) employment at the organisation for less than one year; and (4) a part-time employment contract (less than 32 h per week). Two appointments were planned at the personal office of each participating employee, minimally nine days apart, to collect the movement data. During the measurement period, participants were asked to maintain their daily routines (e.g., not to change anything in their physical activity routines and not to change their use of the sit–stand workstation).

### 2.3. Outcomes

#### 2.3.1. Descriptive Characteristics

Descriptive characteristics were retrieved from the earlier conducted survey study and included age, gender, Body Mass Index (BMI), educational level (MSc or PhD), work experience in years, hours worked per week, means of transportation to and from work (dichotomised into by car or by other means of transportation, including an active component such as walking), and self-reported time based at the personal workstation (dichotomised into <6 h or >6 h). 

Standing episodes at the participant’s sit–stand workstation were assessed for both users and non-users, by using a paper register. This paper register was attached to the personal workstation of the participants, on which they could indicate start and end times of each standing episode. Number and duration of standing episodes were calculated per workday and averaged over all workdays. Furthermore, the distribution in frequency (1−2, 3−4, ≥5 times per day) and duration (<15, 15−30, 30−60, 60−90, ≥90 min per episode) were calculated for average standing episodes. 

#### 2.3.2. Movement Behaviour Outcomes

Movement behaviour outcomes, including sitting time, standing time, stepping time and number of steps, were objectively assessed using the ActivPAL activity tracker (PAL technologies). The ActivPAL is a light-weight device, which has shown to be a reliable instrument to measure and distinguish between sitting time, standing time and stepping time [37,38]. The ActivPAL was worn for seven full consecutive days, on the right mid-thigh, attached with adhesive tape. The ActivPAL was made waterproof by packaging it in a nitrile sleeves, allowing short-term water contact (e.g., taking a shower) but not long-term water contact (e.g., taking a bath). In a diary, participants indicated sleep times, work times and non-wear times (indicated by start and end times). Data were processed using the ProcessingPAL tool, developed by Charlotte Edwardson and Shashidar Ette, which incorporates the algorithm of Winkler et al. (2016) to isolate valid wear time from sleep and non-wear times [39]. If days included <500 steps, ≥95% in any one activity or ≤10 h of waking wear data, these were excluded by the tool. All participants provided data for at least four valid days. Heat maps of wear-time data were created and with an eyeball check, divergent sleeping bouts (i.e., waking or sleeping times very late or early) were compared to sleeping times from the diary. If needed, waking wear times were adjusted. Other movement outcomes were calculated by the tool and included time in static standing bouts (≥15 min, without taking one step in between), number of sit–stand transitions, and time in sitting bouts <30 min and ≥30 min. Furthermore, the tool could isolate all movement outcomes (except for number of steps) for working hours as indicated in the diary. Outcomes for non-working hours were calculated by subtracting outcomes during working hours from workday outcomes. Number of sitting bouts (0−15 min, 15−30 min, 30−45 min and ≥45 min) were calculated from the tool and averaged for working and non-working hours. Mean movement behaviour outcomes for workdays (minimally 3–maximally 5 days) and non-workdays (minimally 1–maximally 3 days) were calculated using the diary data to identify work and non-workdays. For total days (all valid days combined), sitting time, standing time, stepping time and number of steps were also calculated by the tool. Working and non-working hours were normalised to 8 h and workdays, non-workdays and total days were normalised to 16 h. 

### 2.4. Analyses 

Differences between users and non-users in descriptive data were analysed using an independent t-test (for linear outcomes) or chi-square test (for dichotomous outcomes). Descriptive information of self-reported frequencies and duration of standing episodes was obtained for users. Differences between groups in movement behaviour outcomes were analysed using linear regression, including user profile as the independent variable. Only crude models were used, since the two user profiles were intentionally chosen and differences between groups might be unjustifiably corrected if any confounding would be taken into account. Level of statistical significance was set on *p* < 0.05. Static standing bouts appeared to be non-normally distributed and skewed to the right. For this outcome, medians (interquartile range 25−75 (IQR)) were calculated and no statistical test for differences between groups was performed.

## 3. Results

### 3.1. Participants

A total of 103 employees (44 users; 59 non-users) were invited, of whom 64 showed interest in our study. There were 13 office workers who withdrew before measurements, mainly because of unavailability or because they did not want to commit to wearing the ActivPAL device on the leg for seven days. Two users were excluded from the study, one because of experiencing complaints (inducing non-use of the sit–stand workstation) and one because of the use of a different type of active workstation. In total, 49 (48%) employees—24 users and 25 non-users—provided written consent and were included in the analyses. In Table 1, an overview of descriptive characteristics is provided for users and non-users. Although no statistically significant differences were shown between users and non-users, users were more than three years younger, and correspondingly, had almost three years less work experience than non-users.

Table 1 also presents the user’s self-reported frequency and duration of standing episodes at their workstation. On average, the standing option of the workstation was used 2.63 (SD = 1.30) times per workday with an average standing duration of almost one hour (0.94, SD = 0.54 h). As expected, the standing option was not or very rarely used by non-users, with in total, three non-users using it once during the seven day measurement period.

### 3.2. Movement Behaviour Outcomes for Working and Non-Working Hours

In Table 2, the movement behaviour outcomes for working and non-working hours (at workdays) are shown, including differences between users and non-users. During working hours, users of the sit–stand workstations sat less than non-users (β =−1.64; 95% CI = −2.27–−1.01 h/8 h workday). The lower sitting times were almost completely replaced by higher standing times (β = 1.51; 95% CI = 0.92–2.10 h/8 h workday) in users and by a relatively small (non-significant) amount by stepping time. Users spent less occupational sitting time in short bouts (<30 min) compared to non-users, with a significant difference of −0.86 (95% CI= −1.39–−0.33) hour/8 h workday. Users also spent less time in long sitting bouts (≥30 min), with a significant difference of −0.78 (95% CI= −1.56–−0.00) hour/8 h workday. Users accumulated more time in static standing bouts during working hours (median 0.56; IQR = 0.19–1.08 h/8 h workday), compared to non-users (median 0.07; IQR = 0–0.17 h/8 h workday). During non-working hours, no statistically significant differences were found for movement behaviour outcomes between users and non-users.

### 3.3. Number of Sitting Bouts for Working and Non-Working Hours

Figure 1 presents the frequency of sitting bouts of different durations for working and non-working hours for users and non-users. Bouts of 0–15 min were most prevalent during both working and non-working hours and bouts of 45 min or (much) longer were rare. For users, during working hours, the number of short sitting bouts (0–15 min) was relatively low (17.1) compared to non-users. This reflected the lower number of sit-to-stand transitions during working hours for users compared to non-users.

### 3.4. Movement Behaviour Outcomes for Total Days, Workdays and Non-Workdays

In Table 3, the movement behaviour outcomes (sitting, standing, stepping time and number of steps) are shown for total days and for workdays and non-workdays separately, including differences between users and non-users. For total days, statistically significant differences between users and non-users for sitting time and standing time were found, with less sitting (β = −0.95, 95% CI = −1.69–−0.21 h/16 h day) and more standing (β = 0.88, 95% CI = 0.27–1.49 h/16 h day) for users. These differences between users and non-users seemed mostly accounted for by differences in sitting and standing time during workdays, with β = −1.62 (95% CI = −2.46–−0.78) and β = 1.50 (95% CI= 0.78−2.23 h/16 h day), respectively. Time in static standing bouts was for users relatively high during workdays, with a median of 0.66 (IQR = 0.25−1.11 h/16 h day), but lower during non-workdays. Other differences between groups, including all movement behaviour outcomes for non-workdays, were not statistically significant.

## 4. Discussion

This study looked into differences in movement behaviour outcomes between users and non-users of the standing option of their sit–stand workstation, to which all participants have had long-term access. Differences between users and non-users were mainly found during working hours; users sat 1.64 h/8 h workday less and stood 1.51 h/8 h workday more than non-users. In most intervention studies including sit–stand workstations, reductions in sitting time were mainly replaced by increased standing time [40]. Particularly, time in static standing bouts seemed to be high in users in the current study, which was in line with the relatively high self-reported durations of standing episodes (mean 0.94; SD = 0.54 h/episode). For non-working hours and non-workdays, no statistically significant differences were found in movement behaviour outcomes between users and non-users, suggesting there were no major carry-over or compensation effects outside working hours. Over the total days of the measurement period, differences were attenuated but remained substantial between users and non-users.

General sedentary office populations have shown to be more sedentary during workdays (9.7 h) than non-workdays (7.7 h) [35]. This pattern was reflected by the non-users, but not by the users in the current study. Compensatory effects (reduced sitting during working hours, but increased sitting during non-working hours) have been reported after the implementation of sit–stand workstations, but these were group effects at three months of follow-up [41]. In the study of Mazzotta et al (2018), habitual users of sit–stand workstations showed similar amounts of occupational sitting and total sitting time as the users in the current study [11]. Additionally, in long-term follow-up measurements on the effectiveness of an intervention, including sit–stand workstations, reductions in sitting time during working hours attenuated for total sitting time in the intervention group [10]. This indicated that interventions to reduce sitting at work, such as sit–stand workstations, do not seem to carry over to movement behaviours outside working hours and might lead to some compensation behaviour [42]. 

In the current study, overall activity levels for both users and non-users were relatively low, with a total average of around 5200 steps per day. Health risks associated with prolonged sitting have shown to be higher when combined with low levels of moderate to vigorous intensity physical activity [43], which might apply especially to the non-users in the current study. However, despite their higher sitting times, non-users interrupted their sitting during working hours more frequently compared to users. Still, although energy expenditure increases during sit–stand transitions [44], an unrealistic amount of sit-to-stand transitions should be performed to reach meaningful effects [45], which was by far not met by the non-users (n = 144 versus n = 32, respectively). Furthermore, it is unclear what the impact is of the duration of bouts and the frequency of breaks of sitting on health [43]. Nonetheless, replacing prolonged sitting solely with standing is not an optimal solution, since it has been shown that prolonged occupational standing is associated with health risks such as developing varicose veins [46,47]. The relatively long time spend in occupational standing bouts by users might indicate that some of them exchanged one static posture (sitting) for another (standing), with increased risks of detrimental health consequences as well [48]. Within this office population, for both user and non-users, additional interventions aiming at the (correct) use of sit–stand workstations and how to find a healthy balance between sitting, standing and moving seem worthwhile. 

### 4.1. Study Limitations and Strengths

Strengths of this study included the objective measurement of natural movement behaviour outcomes and the comparison of habitual users of sit–stand workstations with non-users, all of whom had long-term access to the furniture. Furthermore, we reported the movement behaviour outcomes during and outside working hours and on workdays and non-workdays. Still, it should be noted that the actual use of the standing option was self-reported, and for this reason, we only descriptively reported it for users. Other limitations of this study include the specific (highly educated) population in which it was conducted. However, despite the high level of education, volumes of sitting times seemed generalizable towards broader office populations as they were comparable to those from earlier studies among more general office populations including user [11] and non-user populations [49]. Still, movement behaviour outcomes might vary between office populations [50]. Furthermore, although we reported on numerous different movement outcomes and distinguished between users and non-users, we reported overall group averages, neglecting possible variation within subjects. 

Outcomes may vary over days within subjects, as shown previously for frequency and duration of sitting and standing episodes [51]. Although the data of seven days might be enough to represent an average for total days, we also distinguished between workdays and non-workdays, which involved a fewer number of days. Although we asked participants to maintain their daily routines during the research period, the measurements might still have (unconsciously) influenced their movement behaviour patterns. For example, the users could have increased the use of the standing option when compared to non-measurement periods. Another limitation was the possible selection bias of including more physically active participants, because they may be more interested in participating in a study involving the measurement of movement behaviour. If we indeed selected relatively active non-users, then this might account for their relatively active non-working hours and non-workdays and caution is needed when interpreting these results. Lastly, since movement behaviour outcomes (sitting, standing and stepping) are in nature co-dependent, compositional data analyses (CoDA) could have been conducted to analyse differences between users and non-users in the current study. This CoDA method has shown to be valuable, for instance, when examining associations between movement behaviour and health outcomes [52,53,54,55]. Furthermore, it has been shown that CoDA and standard analyses of movement behaviour outcomes provided different effect sizes between age and gender groups in blue collar workers, especially when the variation of (occupational) movement behaviour between workers was high [56]. Still, the data from the current study were normalised to 16 h (for total, work and non-workdays) and to 8 h (for working hours and non-working hours). Furthermore, variation of movement behaviour at work within this sample of office workers might have been much smaller than in blue collar workers [56], especially since we categorised both groups based on their movement behaviour (i.e., use of sit–stand workstations), limiting the potential divergence between CoDA and current analyses.

### 4.2. Practical Implications

In the current study, it seemed that habitual users of the standing option did not necessarily have an overall ‘more physically active’ profile compared to non-users, although they sat less and stood more, especially at work. Motivation to adopt the use of the standing option might not correspond to a physical activity profile outside work. Adding motivational interventions at work might provide opportunities for non-users to change their behaviour [57] and adopt the use of the standing option. Providing training [58] and reminders [28,29,30] has shown to be effective to increase the use of sit–stand workstations, but these were not structurally provided to our study population. Still, with adherence to interventions to reduce sitting, static prolonged standing needs to be avoided too [59], and this message could be incorporated in the training and reminders. In both users and non-users, activity levels were low and additional (leisure time) interventions might be needed. For both (potential) users and non-users of sit–stand workstations, it is essential to address the importance to alternate between postures, avoiding prolonged sitting as well as prolonged standing and including movement, such as walking, throughout the day. 

## 5. Conclusions

Habitual users of sit–stand workstations showed significantly lower sitting times and significantly higher standing times during working hours than non-users, who also had long-term access to this furniture. However, during non-working hours and on non-workdays, no differences in movement behaviour outcomes were found between users and non-users. This suggests that the use of sit–stand workstations does not necessarily imply a less sedentary and more physically active lifestyle outside the workplace. For both users and non-users, additional interventions aiming at (correct) use of sit–stand workstations and how to find a healthy balance between sitting, standing and moving could be beneficial.

## Figures and Tables

**Figure 1 ijerph-17-04075-f001:**
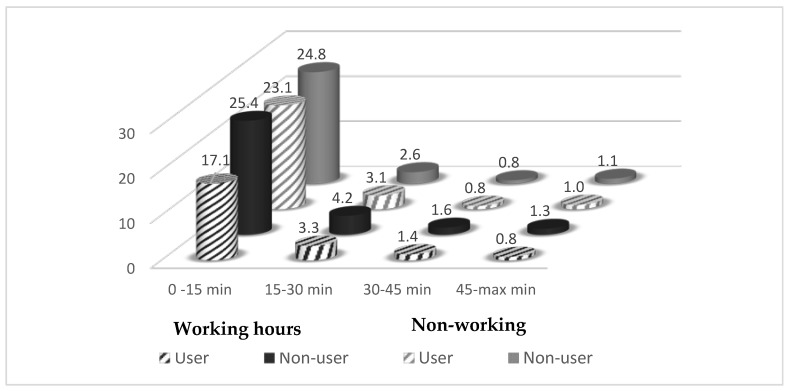
Number of sitting bouts of different durations for users and non-users during working and non-working hours.

**Table 1 ijerph-17-04075-t001:** Descriptive characteristics, including the use of the standing option for users.

Descriptive Characteristics	Users N = 24	Non-Users N = 25	*p*-Value
Gender, N males (%)	19 (79%)	16 (64%)	0.24
Age, mean (SD) years	40.46 (7.04)	43.80 (6.81)	0.10
BMI, mean (SD) kg/m2	23.88 (3.19)	23.78 (3.88)	0.92
Education with PhD degree, N(%)	11 (46%)	13 (52%)	0.67
Work experience, mean (SD) years	10.42 (5.75)	13.16 (8.17)	0.18
Work time, mean hours (SD) / week	38.00 (4.98)	38.74 (3.98)	0.58
Transportation to work by car, N (%)	8 (33%)	9 (36%)	0.85
Based at workstation > 6 h per workday ^1^, N (%)	18 (75%)	24 (96%)	0.05 *
**Self-reported standing episodes at own workstation**			
Frequency ^2^ per day, mean (SD)	2.63 (1.30)	-	
1−2 times per day, N (%)	13 (54%)	-	
3−4 times per day, N (%)	9 (38%)	-	
5 times of more per day, N (%)	2 (8%)	-	
Duration ^2^ per episode, mean (SD) hour	0.94 (0.54)	-	
<15 min, N (%)	-	-	
15–30 min, N (%)	5 (21%)	-	
30–60 min, N (%)	13 (54%)	-	
60–90 min, N (%)	4 (17%)	-	
90 min or longer, N (%)	2 (8%)	-	

* statistically significant difference (*p* < 0.05); ^1^ Because of low numbers, Fisher’s Exact test instead of a chi-square test was performed; ^2^ Frequency and duration were averaged per workday and over total number of workdays.

**Table 2 ijerph-17-04075-t002:** Movement behaviour outcomes for working and non-working hours and differences between users and non-users (β) including 95% confidence intervals (95% CI).

Outcomes in Mean (SD) Per 8 h Workday		Users N = 24	Non-Users N = 25	β (95% CI)
Sitting time, hour	Working hours	3.98 (1.19)	5.62 (1.00)	–1.64 (−2.27–−1.01) *
	Non-working hours	4.11 (0.87)	4.11 (0.71)	−0.01 (−0.46- 0.45)
Standing time, hour	Working hours	3.30 (1.09)	1.79 (0.96)	1.51 (0.92–2.10) *
	Non-working hours	2.58 (0.63)	2.61 (0.49)	−0.03 (−0.36 −0.29)
Stepping time, hour	Working hours	0.73 (0.3)	0.60 (0.21)	0.13 (−0.02–0.28)
	Non-working hours	1.31 (0.45)	1.28 (0.41)	0.04 (−0.21–0.29)
Number of sit-to-stand transitions	Working hours	22.34 (7.8)	32.06 (20.99)	−9.72 (−18.90–−0.54) *
	Non-working hours	28.04 (8.18)	29.64 (8.39)	−1.60 (−6.37–3.17)
Time in sitting bouts <30 min, hour	Working hours	2.27 (0.83)	3.13 (1.00)	−0.86 (−1.39–−0.33) *
	Non-working hours	2.37 (0.59)	2.24 (0.63)	0.13 (−0.22–0.48)
Time in sitting bouts ≥30 min, hour	Working hours	1.7 (0.94)	2.48 (1.65)	−0.78 (−1.56–−0.003)*
	Non-working hours	1.74 (1.00)	1.87 (0.89)	−0.14 (−0.70– 0.41)
Time in static standing bouts ^1^	Working hours	0.56 (0.19−1.08)	0.07 (0–0.17)	
	Non-working hours	0.01 (0−0.10)	0.05 (0–0.12)	

* Statistically significant difference between users and non-users with *p* < 0.05; ^1^ Presented by median (25–75 interquartile range) because of non-normal distribution of the data, with no statistical tests performed.

**Table 3 ijerph-17-04075-t003:** Movement behaviour outcomes for total days, workdays and non-workdays and differences between users and non-users (β), including 95% confidence intervals (95% CI).

Outcomes in Mean (SD) per 16 h day		Users N = 24	Non-Users N = 25	β (95% CI)
Sitting time, hour	Total days	8.25 (1.44)	9.20 (1.13)	−0.95 (−1.69–−0.21) *
	Workdays	8.15 (1.59)	9.77 (1.33)	−1.62 (−2.46–−0.78) *
	Non-workdays	8.47 (1.77)	7.75 (1.59)	0.73 (−0.25–1.70)
Standing time, hour	Total day	5.65 (1.15)	4.77 (0.98)	0.88 (0.27−1.49) *
	Workday	5.87 (1.32)	4.37 (1.20)	1.50 (0.78−2.23) *
	Non-workday	5.17 (1.35)	5.75 (1.27)	–0.57 (–1.34–0.19)
Stepping time, hour	Total day	2.10 (0.64)	2.03 (0.52)	0.07 (–0.26−0.41)
	Workday	1.98 (0.57)	1.86 (0.48)	0.12 (–0.18–0.43)
	Non-workday	2.35 (0.86)	2.51 (0.82)	−0.15 (–0.64–0.34)
Number of steps	Total day	5378 (1812)	5108 (1442)	270 (−669−1210)
	Workday	5220 (1588)	4802 (1372)	417 (−435−1269)
	Non-workday	5645 (2521)	6032 (2390)	−386 (−1813–1041)
Number of sit-to-stand transitions	Workday	50.43 (12.21)	61.37 (25.35)	−10.94 (–22.45−0.58)
	Non-workday	60.22 (26.92)	57.87 (17.86)	2.35 (–10.82−15.53)
Time in sitting bouts <30 min, hour	Workday	4.71 (1.03)	5.36 (1.44)	−0.65 (–1.37−0.08)
	Non-workday	4.41 (1.12)	4.25 (1.30)	0.16 (–0.55–0.87)
Time in sitting bouts ≥30 min, hour	Workday	3.43 (1.63)	4.41 (2.27)	−0.98 (−2.12−0.16)
	Non-workday	4.07 (1.81)	3.50 (1.56)	0.57 (–0.41–1.55)
Time in static standing bouts ^1^	Workday	0.66 (0.25–1.11)	0.16 (0.06–0.36)	
	Non-workday	0.14 (0−0.28)	0 (0–0.37)	

* Statistically significant difference between users and non-users with *p* < 0.05; ^1^ Presented by Median (25–75 interquartile range) with no statistical tests performed because of non-normal distribution of the data.

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
