# Peer review of "Natural Patterns of Sitting, Standing and Stepping During and Outside Work—Differences between Habitual Users and Non-Users of Sit–Stand Workstations"

_ijerph, 2020, doi:10.3390/ijerph17114075_

Round 1

Reviewer 1 Report

Dear authors,

This is an article focus on of an area with good justification and need to go deeper into the topic "Natural patterns of sitting, standing and stepping during and outside work  - differences between habitual users and non-users of sit-stand workstations. Overall most methods have been employed to a good standard and described well. I have a few comments and suggestions to help improve clarity in parts of the paper (attached document).

Best regards.

Author Response

Dear editor,

We would like to thank the editor and reviewers for their feedback and we appreciate the opportunity to resubmit this manuscript. We have addressed all comments from reviewer 1 point-by-point here below and feel it has improved the manuscript.

Reviewer 1

This is an article focus on of an area with good justification and need to go deeper into

the topic Natural patterns of sitting, standing and stepping during and outside work

– differences between habitual users and non-users of sit-stand workstations. Overall

most methods have been employed to a good standard and described well. I have a few

comments and suggestions to help improve clarity in parts of the paper:

  1. In the abstract, lines 17-18: The objective of this study was to examine outcomes between habitual users and non-users of sit-stand workstations, but to examine the results for what purpose?
    Response: Thank you for pointing this out. We have now included the purpose to the objective of this study. The abstract now reads [line: 16-20]: “Sit-stand workstations have shown to reduce sitting time in office workers on a group level. However, movement behaviour patterns might differ between subgroups of workers. Therefore, the objective of this study was to examine sitting, standing and stepping outcomes between habitual users and non-users of sit-stand workstations.”
  2. In the introduction, it would perhaps be good to mention that there are previous systematic reviews that analyze workstations in other diseases in addition to type II diabetes, cardiovascular disease and premature mortality (lines 56-57), for example overweight-obesity and low back discomfort (Other references: 1. Use of Active Workstations in Individuals with Overweight or Obesity: A Systematic Review. Josaphat KJ, Kugathasan TA, E R Reid R, Begon M, Léger PM, Labonté-Lemoyne E, Sénécal S, Arvisais D, Mathieu ME. Obesity (Silver Spring). 2019 Mar;27(3):362-379. doi: 10.1002/oby.22388. Epub 2019 Feb 5. / 2. Sit-stand workstations and impact on low back discomfort: a systematic review and meta-analysis. Agarwal S, Steinmaus C, Harris-Adamson C. Ergonomics. 2018 Apr;61(4):538-552. doi:10.1080/00140139.2017.1402960. Epub 2017 Dec 4. Review).
    Response: Thank you for these additional references. We have now included the references of Josaphat et al. (2019) and Agarwal et al. (2018) in the Introduction to provide a more comprehensive picture [line: 61-63]: “Furthermore, using active workstations have shown to increase energy expenditure in individuals with overweight and obesity (ref Josaphat et al 2019), and the use of sit-stand workstations may reduce the risk of low back pain (ref Agarwal et al 2019).”

  1. Lines 92-93: This study has specified that participants were selected from a previous study, but how was that selection made in this previous study?
    Response: We have clarified that the selection of previous study was a randomly drawn sample [line 97-99]: “Participants were selected from an earlier conducted survey study [36], in which participants (N=1098 from a randomly drawn sample of employees within the organisation) had indicated their frequency of use of the sit-stand workstation.”

  2. Line 44: “…their effect on productivity and their effect on musculoskeletal symptoms”, maybe it seems to be repetitive…their effect and their effect.
    Response: Thank you for pointing this out. We have removed this repetition now [line: 45-46]: “…have increasingly been evaluated on their effectiveness to reduce sitting time [3], their effect on productivity [4, 5] and musculoskeletal symptoms [6].”

  3. Line 54-55: “…research field of sedentary behaviour. In this research field…”. Maybe you could chain the phrase with a comma, instead of repeating research field again.
    Response: Thank you for this suggestion, we have adjusted the phrasing accordingly [line: 55-57]: “…the recently emerging research field of sedentary behaviour, which has shown that prolonged sitting, accumulated during the whole day, seems to be associated with several health risks…”

  4. Line 253 and 266: “…in our study”…perhaps it is better to write “in the present study or in the current study or in this study”
    Response: Thank you for this suggestion. We have rewritten both phrases as suggested:
  5. -line 256-257: “This pattern was reflected by the non-users, but not by the users in the current study.”
    -
    line 269-270: low levels of moderate to vigorous intensity physical activity [40], which might apply especially to the non-users in the current study.”

Reviewer 2 Report

This manuscript is written clearly, with methods and results described well. I only have a few questions/comments:

1) In the introduction, page 2 paragraph 3 (line 74-76), reference 33 (Clemes et al) is said to provide evidence that "office workers without access to sitting workstations" are more sedentary during work days/hours. This reference however describes its study population as a standard office worker population, with no comment made about sitting or sit-stand workstations. Perhaps this is an error - either the reference was not the intended one, or the sentence needs editing.

2) Methods page 4 paragraph 2 (line 150-151) the number of workdays and non-workdays indicates that some participants only were required to provide 6 days of data (e.g. someone reporting a minimum of 3 workdays + maximum of 3 non-workdays = 6 workdays, not 7 as the protocol stated). Earlier in the methods (page 3, line 94) states that data were collected for a full 7 days per person. Just checking if this was intentional and perhaps reflects the usable data provided by participants (where some days were excluded by the ProcessingPAL analysis tool, or if there was an error in that paragraph on page 4.

3) A statistician could be consulted regarding suitable analysis for non-normally distributed categorical data in table 1 and table 2. E.g. Table 1 could use a Fisher's Exact Test where the Chi-square is not appropriate due to small numbers in each group. 

4) Discussion line 2, suggest "users of the standing desk option" (i.e. suggest inclusion of the word 'desk' here). Discussion line 3, at the end of this line, 'user' should be plural: 'users'.

Author Response

Please see atached

Reviewer 3 Report

I found this is a very timely study that shows significant difference between users’ and non-users’ sitting/standing duration during working hours. The research design and measurement methods were appropriate. The manuscript was easy to follow.

A few minor suggestions:

  1. Definition of “User” and “Non-user”:

The authors defined “user” as “using it at least daily” and “non-user” as “using it less than once per week.” What is “using it?” Wouldn’t it be “using it at a standing position?”

  1. Page 3, Line 121. A “paper register” was used to document self-report standing episodes. I am a little concerned about the reliability of the self-report data. Could the authors provide a reliability test result for this tool? I believe it could be easily done by coupling the ActivPAL data.

  1. Page 3, Line 107. Why “32 hours per week” was used as a cut-point?

  1. Page 4, Lone 175. What does the “activity permissive workstation” mean?

  1. For the consent form/informed consent, were the participants informed the aims of this study? Would it be possible that the “users” intentionally used the sit-stand workstation more frequently if they already knew the study aim?

  1. For the Background/literature review, a recently published literature review may be helpful:
    Zhu, X., Yoshikawa, A., Qiu, L., Lu, Z., Lee, C., & Ory, M. (2019). Healthy workplaces, active employees: A systematic literature review on impacts of workplace environments on employees’ physical activity and sedentary behavior. Building and Environment, 106455. doi:10.1016/j.buildenv.2019.106455
